# Determinants and Sources of Iron Intakes of Australian Toddlers: Findings from the SMILE Cohort Study

**DOI:** 10.3390/ijerph16020181

**Published:** 2019-01-09

**Authors:** Jane A Scott, Georgina Gee, Gemma Devenish, Diep Ha, Loc Do

**Affiliations:** 1School of Public Health, Curtin University, Perth, WA 6102, Australia; georgina.gee@postgrad.curtin.edu.au (G.G.); gemma.devenish@curtin.edu.au (G.D.); 2Australian Research Centre for Population Oral Health, The University of Adelaide, Adelaide, SA 5000, Australia; diep.ha@adelaide.edu.au (D.H.); loc.do@adelaide.edu.au (L.D.)

**Keywords:** iron intake, determinants, food sources, toddlers

## Abstract

The first two years of life is a period of rapid growth and development. During this time a lack of key nutrients, including iron, can have long-lasting effects on motor and cognitive performance. The purpose of this cross-sectional study was to determine intake and sources of iron in a cohort of 828 toddlers (mean age; 13.1 mo) participating in the Adelaide-based Study of Mothers’ and Infants’ Life Events affecting oral health (SMILE), and to identify determinants of iron intake. At approximately 12 months of age, 3 non-consecutive days of dietary intake data were collected using a 24-h recall and 2-days food record. The Multiple Source Method was used to combine data from the 24-h recall and each day of the food record to estimate usual iron intake and descriptive statistics were used to report sources of iron. Linear regression was used to identify associations between iron intake and non-dietary determinants (maternal age, education, country of birth, BMI, socioeconomic position, parity, toddler sex) and primary milk feeding method at 12 months. The mean intake of iron was 7.0 (95% CI 6.7–7.2) mg/day and 18.2% of children had usual intakes below the estimated average requirement of 4 mg/day. The main sources of iron included infant and toddler cereals and formulas. Milk feeding method and parity were significantly associated with iron intake. Toddlers with siblings and those who received breast milk as their primary milk feed had significantly lower iron intakes than only children and those who received formula, respectively. The Australian Infant Feeding Guidelines promote the importance of iron-iron-rich complementary foods such as meat and meat alternatives. However, low intakes of this food group suggest that parents do not recognize the importance of these foods or understand the specific foods that toddlers should be eating.

## 1. Introduction

The first 1000 days of life—which includes gestation and the first two years of life-is a period of rapid growth and development. This time span presents a unique window of nutritional opportunity, as it is the period when the foundations of optimum health, growth, and development across the lifespan are established. Conversely, it is also a time of great vulnerability [1,2,3]. While all organ systems undergo rapid growth and development during the first two years of life, it is a particularly important period for neurodevelopment. During this time a lack of key nutrients, including iron, can have long-lasting effects on motor and cognitive performance [4], and subsequently on adult human capital and economic productivity [1,3].

Iron deficiency (ID) is the most prevalent micronutrient deficiency in children [4] and when left untreated may progress to the more serious condition, iron deficiency anemia (IDA) [5]. In high income countries such as Australia, ID and IDA in toddlers are usually the result of inadequate iron intake or excess intake of cow’s milk [5,6,7].

Despite the recognized importance of the first 1000 days of life, relatively little is known about the nutritional status of Australian children under two years of age. The most recent population-based nutrition and physical activity surveys conducted in Australia did not investigate the diets of children younger than two years [8,9]. Therefore, what is known about the diets in general, and the iron intake in particular, of this age group of Australian children comes from a relatively small number of single-center or regional studies [10,11,12,13,14], or is extrapolated from international studies [5,15,16,17]. The aims of this study were to identify (1) the sources of iron in the diets of a population-based cohort of Australian toddlers aged 12 to 14 months, and (2) the predictors of usual iron intake in this cohort.

## 2. Materials and Methods

### 2.1. Setting and Recruitment

This study is a cross-sectional, secondary analysis of dietary data collected in the Study of Mothers’ and Infants’ Life Events affecting oral health (SMILE). This birth cohort study aims to identify early life events and risk factors, including diet and early childhood feeding practices, associated with early childhood caries and obesity [18]. SMILE was originally designed to follow children from birth into their third year of life, but the project has received additional funding to follow the cohort until 7 years of age. Between July 2013 and August 2014, 2147 mothers and 2181 newborns, including 34 sets of twins, were recruited from three major maternity hospitals servicing Adelaide in South Australia. All new mothers with sufficient English competency were invited to participate with the exception of those mothers intending to move out of the greater Adelaide area within a year. Mothers delivering in hospitals which service lower socioeconomic areas were oversampled in order to compensate for anticipated higher attrition rates [18]. The study was approved by the Southern Adelaide Clinical Human Research Ethics Committee (HREC/50.13, approval date: 28 February 2013) and the South Australian Women and Children Health Network (HREC/13/WCHN/69, approval date: 7 August 2013). Signed informed consent was obtained from all mothers.

### 2.2. Collection and Handling of Dietary Data

Three days of dietary data were collected using a combination of a single 24-h recall and 2 non-consecutive days of estimated food records (FR). Between July 2014 and August 2015 when children reached 12 months of age, FR booklets and a letter advising of the impending 24 h recall were mailed to the 1921 mothers remaining in the study. The FR booklets included detailed instructions for completion, consisting of an example of a completed one-day FR and images of portion sizes and common household measures to assist with portion estimations. The 24 h recalls were conducted by telephone using the five-step multi-pass method [19] by one of two trained dietitians who referred to the FR booklet images to assist with quantifying portion sizes. At the end of the interview two non-consecutive days (one weekday and one weekend day) were allocated for the FR.

Dietary intakes were entered into Foodworks version 8 (Xyris Software (Australia) Pty Ltd., Brisbane, Australia) for analysis using the AUSNUT 2011-13 food composition database [20]. Data were double-entered by trained nutritionists/dietitians, using data entry protocols and calibration procedures for standardization. Nutrient data for 187 commercial infant food products not found in this food composition database were added to the database as new foods using information from the product’s nutrition information panel or the manufacturer’s website, mapped to a similar product in AUSNUT 2011–13 for missing micronutrient values. Each new food was assigned an 8-digit food code, following the AUSNUT naming conventions. Breast milk intake was estimated using the method employed for this age group in the UK 2011 Diet and Nutrition Survey of Infants and Young Children [16]. Breastfeeds were recorded in minutes and the amount of milk consumed was calculated as 10 g/min to a maximum of 100 g per feed, as the contribution to nutrient intake after 10 min of breastfeeding is considered minimal in this age group [21].

### 2.3. Statistical Analysis

Data from Foodworks were exported to Microsoft Access (Microsoft Office 2016, Albuquerque, NM, USA), then imported to SPSS Statistics for Windows, version 24.0 (IBM Corp, Armonk, NY, USA) for statistical analysis. To account for intra-individual variability across consumption days, the Multiple Source Method (MSM) was used to combine data from the 24-h recall and each day of the food record to calculate usual daily iron intake for each participant employing free-to-use software developed for use in the European Food Consumption validation project [22,23]. The MSM is a multistep method which aims to estimate usual food intake distributions by estimating a consumption probability (step 1) and a consumption-day amount (step 2). An estimate of an individual’s usual intake is then obtained by multiplying consumption probability and consumption-day amount (step 3) [23]. The method can be used also to estimate the usual intake of nutrients that are consumed daily by using only the consumption-day amount part of the model [24], as was done in this analysis. The mean and 95% confidence interval (CI) of usual iron intake, and the proportion of children with usual iron intakes below the Australian estimated average requirement (EAR) for this age group of 4 mg/day [25] was calculated for the whole sample and by socio-demographic factors.

To identify important sources of iron, the 2303 individual foods consumed by children were grouped into food groups and subgroups using the standard food groupings in the AUSNUT 2011-13 food coding system. The mean, standard deviation, median and quartiles contribution of iron, and the percentage contribution of each food group to total iron intake were calculated for the whole sample and for consumers of each food group. For consumers, the mean percentage of the recommended dietary intake (RDI) for this age group of 9 mg/day [25] derived from each food group was calculated.

Explanatory variables investigated as potential predictors of iron intake included maternal age at baseline (<25 y, 25–34 y, and ≥35 y); highest level of maternal education (high school/vocational or some university and above); maternal country of birth (Australia and New Zealand, India, China, Asia-Other, United Kingdom (UK) and Other); maternal pre-pregnancy body mass index (BMI) kg/m^2^ (>25, 25–29.99 and ≥30); parity (primiparous and multiparous) and child’s sex. Residential postcodes were used to derive Index of Relative Socio-Economic Advantage and Disadvantage (IRSAD) deciles, with decile 1 being most disadvantaged and decile 10 being least disadvantaged [26]. These deciles were collapsed into 5 groups (deciles 1–2, deciles 3–4, deciles 5–6, deciles 7–8, and deciles 9–10). Current milk-feeding method was categorized according to the primary source of ‘milk’ consumed as ‘breast milk’, ‘formula’, ‘mixed’ (both breast milk and formula) and ‘neither’ (neither breast milk nor formula). Children receiving breast milk and/or formula may have consumed small amounts of animal or plant-based milks.

The General Linear Model (GLM) procedure in SPSS was used to run multiple linear regression analysis for investigating the association between explanatory factors and usual iron intake. Factors that were significantly associated with iron intake (*p* < 0.05) in the simple linear regression analyses were simultaneously entered into the multiple linear regression model to identify independent associations. The distributions of the outcome variables were slightly skewed, therefore data were analyzed using both parametric and non-parametric analyses. As the results for both analyses were similar, the effects of the independent variables from the parametric analyses are presented for ease of interpretation. Results are presented as the unadjusted and adjusted mean usual iron intake, with 95% CI and p values obtained from regression analyses. For all statistical analyses, a *p*-value of <0.05 was considered statistically significant.

Sensitivity analysis was undertaken to account for extreme over- and under-reporting [27]. As the child’s current weight was unknown a plausible energy intake was estimated using a sex specific estimated energy requirement (EER) for a reference child of the participant’s age [25]. The degree of under- and over-reporting was estimated by calculating the ratio of reported energy intake (EI) to the EER for each child. Children with a ratio of EI:EER below 0.54 or above 1.46 were deemed to have implausible intakes [10]. Primary analyses were performed on the whole sample, and then repeated with participants with plausible energy intakes to test the robustness of the findings. The MSM was applied to the data for these two groups separately.

## 3. Results

### 3.1. Participant Characteristics

Of the 1921 mothers sent food records, 847 (44.1%) completed and returned food records, and 1165 (60.7%) completed the 24 h recall interview. Three days of useable dietary data were available for 828 (43.1%) children, and of these 703 (84.9%) had a plausible energy intake (Appendix B). The majority of participant mothers were 25–34 years of age (69.3%), had commenced or completed university studies (56.5%) and were born in Australia (73.7%) (Table 1). The mean age of children included in this analysis was 13.1 (SD 0.8) months, and 54.6% were male.

### 3.2. Iron Intake

The mean usual daily iron intake for all participants was 7.0 (95% CI 6.7–7.2) mg, and 6.7 (95% CI 6.5–6.9) mg for those with plausible energy intakes. Only 24.0% (*n* = 199) of all children had usual intakes which met or exceeded the RDI of 9 mg/day and 18.2% (*n* = 151) of all children had usual intakes below the EAR of 4 mg/day. One in five children (20.6%) with plausible energy intakes had usual iron intakes below the EAR.

### 3.3. Sources of Iron Intake

The main contributors to iron intake for the whole cohort (Table 2) included infant and toddler formulas (29.6%); commercial infant and toddler food products (16.3%), of which infant and toddler cereals made the largest contribution (11.3%); and other cereals and cereal products (25.4%), of which ready-to-eat breakfast cereal made the largest contribution (15.9%). In comparison, breast milk and cow’s milk contributed 0.5% and 1.0% of total iron intake, respectively.

For those who consumed them, infant and toddler formulas contributed to almost half of the RDI for iron, while breast milk and cow’s milk contributed to 1.1% and 1.0% of the RDI, respectively. Other major contributors to the RDI for consumers included infant and toddler cereals (37.6%) and ready-to-eat breakfast cereal (20.4%). Although some form of meat was consumed by the majority (82.6%) of participants, the contribution from the cumulative meat group to the RDI for consumers was only 5.4%.

### 3.4. Determinants of Iron Intake

In the simple (unadjusted) linear regression analyses, there was a significant association between usual daily iron intake and mother’s country of birth (*p* = 0.005). Children born to multiparous women had lower iron intakes than children born to primiparous women (*p* < 0.001) (Table 3). The strongest unadjusted association was with milk-feeding method at 12 months of age (*p* = 0.002), and children primarily fed formula had significantly higher usual iron intakes than those who primarily consumed breast milk only or neither breast milk nor formula (*p* < 0.001). When all significant variables were simultaneously entered into the multiple (adjusted) linear regression model, parity (*p* = 0.096) and mother’s country of birth (*p* = 0.249) were no longer significant. Only milk-feeding method remained significantly associated with usual iron intake (*p* < 0.001), with children who received breast milk only as their primary milk feed having significantly lower iron intakes than children in all other milk feeding groups.

### 3.5. Sensitivity Analysis

Removal of 125 participants with implausible energy intakes resulted in similar findings to the primary analysis (Appendix A), with one exception. Usual iron intake of those children with plausible energy intakes remained independently associated with parity, with children born to multiparous mothers having significantly lower iron intakes than those born to primiparous mothers (*p* = 0.009). Again, primary milk-feeding method was the strongest independent predictor of usual iron intake, with children who received breast milk only as their primary milk feed having significantly lower iron intakes than children in all other milk feeding groups (*p* < 0.001).

## 4. Discussion

This study investigated iron intake, sources of iron and the predictors of iron intake in a cohort of Australian toddlers as they transitioned in their second year of life to the family diet. While it is recommended that children be breastfed for the first 2 years of life [28], the expanding energy and nutrient needs of the toddler requires that breast milk, or infant formula, be complemented with nutrient-dense family foods, with one of the most problematic nutrients being iron [29]. Roughly one in every five children in this study had a usual iron intake below the EAR for iron, potentially placing them at risk of developing ID.

The findings of this study are comparable to those of other Australian studies of older toddlers, including the Childhood Asthma Prevention Study in Sydney [12] which reported the mean iron intake of 429 toddlers (mean age 18.6 months) to be 5.8 (SE 0.23) mg per day, with 23.3% having iron intakes below the EAR. The Melbourne Infant Feeding, Activity and Nutrition Trial (InFANT) reported the mean iron intake of 423 toddlers (mean age 19.6 months) to be 6.6 (SD 2.4) mg per day, with 18.6% found to have inadequate iron intake [14]. When compared to international studies, the mean intake reported in this study is similar to a mean intake of 6.8 (SD 2.6) mg reported in a recent study of 2-year-old Irish toddlers [17] but less than the mean intake of 10 (SE 0.2) mg reported for a cohort of US toddlers aged 12–23.9 months participating in the 2016 wave of the Feeding Infants and Toddlers Study (FITS) [30].

Directly comparing the mean iron intakes between this and other studies is complicated by the fact that the children in other national and international studies were on average 6 to 12 months older than the SMILE cohort, and therefore would be consuming larger volumes of food and have correspondingly higher iron intakes. Similarly, comparing the adequacy of the iron intake of toddlers between countries on the basis of EAR is problematic because of differences in the nutrient reference values used to assess intake in this age group. For instance, the EAR of 3 mg/day proposed by the US Institute of Medicine [31] is lower than the Australian EAR of 4 mg/day [25] and the EAR of 5.3 mg/day proposed by the UK Committee on Medical Aspects of Food Policy [32]. We have previously reported that only 8.3% of children in the SMILE cohort had iron intakes below the age-specific USA EAR for iron compared with 18.2% with intakes below the Australian EAR [13].

Milk feeding method was strongly associated with usual iron intake with children who received formula as their primary milk feed, either alone or in combination with breast milk, having significantly higher usual iron intakes and being less likely to have intakes below the EAR. Consistent with other Australian [14] and international [17] studies, infant or toddler formula was a major contributor of iron in the diets of the SMILE toddlers who consumed formula. This is to be expected, as in Australia infant and follow-on formulas are required by law to be fortified with between 0.2 and 0.5 mg iron/100 kJ or roughly 5 to 16 mg/L [33], while the concentration of iron in both breast milk and cow’s milk is approximately 0.3 mg/L [20]. Just over one-third of children consumed formula on at least one or more of the three days investigated, which is comparable to 32% of children aged 12 to 16 months who were reported to consume formula in an earlier multi-center Australian study [11]. Formula contributed one-third of the overall iron intake for the whole SMILE cohort and half of the RDI for iron in those who consumed it.

The contribution of formula to the diets of children decreases with age when it is replaced by other foods and beverages. For instance, in the InFANT study, formula was the main source of iron (43.5%) in the diets of infants at 9 months but by 20 months of age formula contributed only 8.6% of total iron [14]. Similarly, in the USA 2008 FITS, infant formula was consumed by 75% of infants aged 9–11.9 months and provided 33.7% of total iron intake in the diets [15] but by 15–18 months the proportion of toddlers consuming formula had dropped to 5.1%, with cow’s milk being the most popularly consumed milk [34]. An analysis of data from the 2005–2012 US National Health and Nutrition Examination Survey (NHANES) identified that formula provided only 4.7% of total iron intake in the diets of toddlers aged 12–23.9 months [35]. Toddler formulas are freely and heavily advertised in Australia [36], and globally the sales of toddler formula marketed for children 13–36 months increased by 53.3% between 2008 to 2013, with continued growth of 33.0% projected for the period 2014 to 2018 [37]. Therefore, the contribution of these formula to the iron intake of Australian toddlers is likely to increase as popularity and sales of these formulas increase.

The Australian Infant Feeding Guidelines (IFG) make strong and specific recommendations regarding the introduction of iron-rich complementary foods, and identify iron-fortified cereals and meat as being particularly good sources of iron [38]. Consistent with international studies [35,39], grain and cereal products, specifically ready-to-eat breakfast cereals and infant and toddler cereals, were the highest contributors to both the total iron intake of the whole sample and the RDI for consumers. This was to be expected, as these products are iron fortified, however the bioavailability of iron from other cereal-based products is low [40], so their contribution to iron status may not be substantial [39].

In comparison, red meat, is a rich source of iron, with high bioavailability [40]. However, while just over eight in every ten children consumed some form of animal flesh, less than half consumed red meat, and the contribution from meats to both total iron intake and the RDI for consumers was relatively low. The findings of this study indicate that whilst most children consumed some form of meat across the three days, intake was likely to be irregular, in small amounts and made up of lower iron containing options. Similar findings have been reported by Byrne et al. [11], who identified that almost 50% of toddlers consumed less than 30 g per day from the meat and meat alternatives food group, with lower iron options including eggs, chicken and ham being the most popular items. An earlier study of Australian toddlers [12] reported similar results with a mean intake of meat and poultry products being 32 g per day, and the most popular item being chicken breast.

In the unadjusted analysis for the whole sample, and in the adjusted sensitivity analysis of those with plausible energy intakes, children born to multiparous women had significantly lower iron intakes than those born to primiparous women. This finding is consistent with those of other Australian [41] and international [42] studies which have reported that having a larger household may negatively influence the quality of food offered to young children. This may be the result of time and financial constraints associated with larger families which make it difficult for caregivers to prepare nutritious family meals [42].

Although children who received breast milk as their primary milk feed were more likely to have usual intakes below the EAR, breastfeeding to 12 months and beyond has been proven to provide numerous health benefits to the child and mother [43,44]. As such, it should continue to be championed and strongly promoted. However, it appears that other important messages about infant and toddler feeding are being missed by parents, particularly those relating to iron. Dwyer [45] on reviewing the findings of the 2016 wave of the US FITS study suggested that stronger recommendations are needed so that parents understand “the specific foods children should be eating and the developmentally appropriate times to introduce complementary foods and beverages” (p1578S). This appears also to be the case for Australian parents, particularly with regard to the introduction of iron-rich foods. While the Australian IFG make specific and strong recommendations with regards the introduction of iron-rich foods as first foods [38], the message does not appear to be getting through to parents.

A key limitation in this study was the use of parent-reported measures, which may be susceptible to social desirability bias and misreporting. This may also be exacerbated by difficulties in quantifying portions, given that toddlers eat small volumes of food and meal times can be a messy experience, with much of the food not being ingested [46]. The volume of breast milk consumed was estimated based on the duration of feeding episodes and intake may therefore be under or overestimated. Nevertheless, in either case the iron content of breast milk is extremely low [37] and this is unlikely to have had a marked effect on the estimated iron intake. Although dietary data were returned by less than half the cohort, intentional oversampling of mother-infant dyads from socially disadvantaged areas [18] means that the analysis population consisted of a relatively socio-economically diverse cohort of children and was representative of the population from which it was drawn [13].

## 5. Conclusions

This study confirms the finding of the limited existing research related to the iron intake of Australian children under the age of two years. Nearly one in five children in this study had iron intakes below the EAR, potentially placing them at risk of developing ID and IDA. Infant and toddler formulas were major sources of iron, and children who received breast milk only as their primary milk feed had significantly lower iron intakes than those who received formula. As toddlerhood is an important period of growth and development, it is necessary to ensure that parents of toddlers are educated as to the importance of iron-rich foods in their children’s diets, and this is particularly important for those who continue breastfeeding into the second year of life. Strategies to increase iron intake during this critical stage of development should be trialed and evaluated.

## Figures and Tables

**Table 1 ijerph-16-00181-t001:** Maternal and child characteristics, SMILE study, Adelaide, South Australia.

Characteristic	*n*	%
Maternal characteristics		
Maternal age at recruitment (years)		
<25	73	8.8
25–34	574	69.3
≥35	179	21.6
Not reported	2	0.2
Maternal education completed		
High school/vocational	356	43.0
Some university and above	468	56.5
Not reported	4	0.5
IRSAD score ^(a)^		
Deciles 1–2	120	14.5
Deciles 3–4	173	20.9
Deciles 5–6	174	21.0
Deciles 7–8	160	19.3
Deciles 9–10	195	23.6
Not reported	6	0.7
Maternal country of birth		
Australia and New Zealand	610	73.7
India	50	6.0
UK	31	3.7
China	37	4.5
Asia-Other	52	6.3
All other countries	43	5.2
Not reported	5	0.6
Maternal BMI ^(b)^ (kg/m^2^)		
<25	477	57.6
25–29.99	167	20.2
≥30	140	16.9
Not reported	44	5.3
Parity		
Primiparous	389	47.0
Multiparous	412	49.8
Not reported	27	3.3
Child characteristics		
Child sex		
Male	452	54.6
Female	376	45.4
Not reported	0	0
Primary milk feeding method at 12 months		
Breast milk	218	26.3
Mixed-breast milk and formula	68	8.2
Formula	309	37.3
Neither breast milk nor formula	226	27.5
Not reported	7	0.8

^(a)^ IRSAD, Index of Relative Socio-Economic Advantage and Disadvantage, where decile 1 = most disadvantaged and decile 10 = most advantaged. ^(b)^ BMI: Body Mass Index.

**Table 2 ijerph-16-00181-t002:** Contribution of iron from food groups to total iron intake for all participants (*n* = 828) and percentage contribution of iron from food groups to Recommended Dietary Intake (RDI) for consumers of individual food groups.

Food Group		All Participants					Consumers Only					
		Mean (SD) (mg/day)	Median (mg/day)	25%	75%	% Iron Intake	% Total group	Mean (SD) (mg/day)	Median (mg/day)	25%	75%	% RDI
Total		6.96 (3.32)	6.40	4.45	8.89	-	-	-	-	-	-	-
Breast milk		0.03 (0.06)	0.00	0.00	0.06	0.5	34.6	0.10 (0.06)	0.09	0.06	0.13	1.1
Infant/toddler formula		2.03 (2.81)	0.00	0.00	4.13	29.6	46.0	4.43 (2.56)	4.28	2.49	6.01	49.3
Infant/toddler commercial products		1.12 (2.48)	0.18	0.02	0.92	16.3	78.0	1.44 (2.73)	0.43	0.09	1.28	16.0
	Infant/toddler cereals	0.78 (2.37)	0.00	0.00	0.00	11.3	23.1	3.38 (3.95)	2.11	0.46	4.77	37.6
	Infant/toddler snack food ^(a)^	0.21 (0.47)	0.05	0.00	0.17	3.1	70.0	0.31 (0.54)	0.10	0.05	0.28	3.4
	Infant/toddler savory dishes	0.12 (0.26)	0.00	0.00	0.14	1.8	28.1	0.44 (0.33)	0.35	0.21	0.58	4.9
Cereals and cereal products		1.75 (1.44)	1.45	0.60	2.57	25.4	98.1	1.79 (1.43)	1.50	0.67	2.59	19.9
	Flours, grains	0.07 (0.19)	0.00	0.00	0.03	1.1	48.2	0.15 (0.25)	0.03	0.01	0.18	1.7
	Regular bread, bread rolls	0.38 (0.40)	0.27	0.06	0.58	5.6	79.7	0.48 (0.39)	0.39	0.18	0.66	5.3
	English muffins, flat breads, savory or sweet breads	0.08 (0.21)	0.00	0.00	0.03	1.2	26.4	0.31 (0.31)	0.22	0.12	0.43	3.4
	Pasta (without sauce)	0.07 (0.14)	0.00	0.00	0.08	1.0	40.1	0.17 (0.17)	0.12	0.06	0.23	1.9
	Breakfast cereals–ready-to-eat	1.09 (1.35)	0.75	0.00	1.79	15.9	59.9	1.83 (1.31)	1.59	0.79	2.38	20.4
	Breakfast cereals-porridge style	0.05 (0.17)	0.00	0.00	0.00	0.7	15.7	0.32 (0.33)	0.26	0.10	0.41	3.6
Other cereal-based products and dishes ^(b)^		0.26 (0.34)	0.14	0.03	0.36	3.8	81.6	0.32 (0.35)	0.20	0.09	0.44	3.6
Meat		0.40 (0.44)	0.28	0.06	0.59	5.8	82.6	0.49 (0.44)	0.37	0.18	0.67	5.4
	Red meat ^(c)^	0.21 (0.34)	0.02	0.00	0.31	3.1	51.5	0.41 (0.38)	0.31	0.15	0.56	4.6
	Poultry ^(d)^	0.09 (0.15)	0.03	0.00	0.12	1.3	56.5	0.16 (0.18)	0.10	0.05	0.22	1.8
	Fish and seafood ^(e)^	0.08 (0.20)	0.00	0.00	0.07	1.2	32.8	0.26 (0.28)	0.16	0.07	0.33	2.9
	Processed meats ^(f)^	0.10 (0.22)	0.00	0.00	0.08	1.4	39.4	0.25 (0.28)	0.17	0.05	0.37	2.8
Dairy		0.12 (0.14)	0.09	0.03	0.17	1.8	91.9	0.13 (0.15)	0.10	0.04	0.18	1.5
	Cow’s milk	0.07 (0.08)	0.03	0.00	0.11	1.0	77.4	0.09 (0.08)	0.06	0.02	0.13	1.0
	Yoghurt	0.02 (0.05)	0.00	0.00	0.03	0.4	45.2	0.05 (0.06)	0.04	0.02	0.07	0.6
	Other dairy products ^(g)^	0.03 (0.10)	0.01	0.00	0.03	0.4	70.9	0.04 (0.12)	0.02	0.01	0.04	0.5
Dairy and meat substitutes		0.02 (0.17)	0.00	0.00	0.00	0.4	6.2	0.40 (0.58)	0.14	0.06	0.55	4.4
	Dairy substitutes ^(h)^	0.02 (0.13)	0.03	0.00	0.00	0.2	4.6	0.33 (0.53)	0.10	0.03	0.49	3.6
	Meat substitutes	0.01 (0.11)	0.00	0.00	0.00	0.1	1.7	0.55 (0.67)	0.43	0.12	0.68	6.2
Fruit		0.33 (0.27)	0.26	0.13	0.47	4.8	95.3	0.35 (0.26)	0.28	0.15	0.49	3.9
Vegetables		0.42 (0.39)	0.33	0.15	0.59	6.2	94.3	0.45 (0.38)	0.36	0.18	0.61	5.0
Legumes and pulses		0.09 (0.26)	0.00	0.00	0.00	1.3	20.1	0.45 (0.43)	0.31	0.15	0.61	5.0
Eggs		0.13 (0.26)	0.00	0.00	0.15	1.9	35.2	0.37 (0.32)	0.27	0.13	0.52	4.1
Nuts and seeds		0.02 (0.09)	0.00	0.00	0.00	0.4	24.6	0.10 (0.15)	0.05	0.03	0.11	1.1
Other ^(i)^		0.14 (0.24)	0.06	0.01	0.18	2.0	88.7	0.16 (0.25)	0.08	0.03	0.20	1.7

^(a)^ Infant/toddler snack food-Infant rusks, cereal based snacks; sweet snacks; infant fruit; infant yoghurts and custards; infant fruit gels and vegetable pouches. ^(b)^ Other cereal-based products and dishes-sweet and savory biscuits; cakes, muffins, scones; pastries; batter based products; and mixed dishes where cereal is the major ingredient. ^(c)^ Red meat and dishes-Flesh from beef, sheep, pig and mammalian game; organ meats and offal; mixed dishes where red meat is the major ingredient. ^(d)^ Poultry and dishes-Flesh from poultry and feathered game; mixed dishes where poultry is the major ingredient. ^(e)^ Seafood and dishes-Flesh from fish and seafood; mixed dishes where fish or seafood is the major ingredient. ^(f)^ Processed meats and dishes-Sausages, frankfurts and saveloys; processed meat products; mixed dishes where sausage or processed meat are the major ingredient. ^(g)^ Other dairy products and mixed dishes–Cheese; cream; flavored milk; mixed dishes where milk or milk products are the major ingredient. ^(h)^ Dairy substitutes–Soy based beverages, yogurts and confections; cheese substitutes. ^(i)^ Other–Fats and oils; beverages; soups; snack foods and confectionary; sugar products; condiments, sauces and spreads; special dietary foods; miscellaneous food items (yeast extracts; herbs, spices and seasoning; cooking ingredients).

**Table 3 ijerph-16-00181-t003:** Factors associated with usual iron intakes (mean values and 95% confidence interval) of toddlers (*n* = 828).

Variables	% Below EAR ^(a)^	Unadjusted Mean (mg/day)	95% CI	*p*	Adjusted Mean (mg/day)	95% CI	*p*
**Total sample**	18.2	7.0	6.7–7.2				
**Maternal characteristics**							
Maternal age at recruitment (years)				0.689			
<25	16.4	6.7	5.9–7.4				
25–34	17.8	7.0	6.8–7.3				
≥35	20.1	6.9	6.5–7.4				
Maternal education-highest level completed				0.489			
High school/vocational	16.9	7.1	6.7–7.4				
Some university and above	19.2	6.9	6.6–7.2				
IRSAD ^(b)^ score				0.924			
Deciles 1–2	19.2	6.9	6.3–7.4				
Deciles 3–4	19.1	6.8	6.3–7.3				
Deciles 5–6	18.4	7.0	6.5–7.5				
Deciles 7–8	19.4	7.1	6.6–7.6				
Deciles 9–10	15.4	7.1	6.6–7.5				
Maternal country of birth				0.005			0.249
Australia and New Zealand	17.2	6.9	6.6–7.1		6.4	6.1–6.7	
India	22.0	6.8	5.9–7.7		6.6	5.9–7.4	
China	8.1	8.2	7.1–9.3		6.4	5.5–7.4	
UK	16.1	6.7	5.6–7.9		6.1	5.0–7.2	
Asia Other	28.8	8.4	7.5–9.3		7.7	6.9–8.5	
Other	20.9	6.5	5.6–7.5		6.9	6.0–7.8	
Maternal BMI ^(c)^ (kg/m^2^)				0.765			
<25	19.1	6.9	6.6–7.2				
25–29.99	19.8	7.0	6.4–7.5				
≥30	15.0	7.1	6.6–7.7				
Parity				0.002			0.096
Primiparous	14.7	7.4	7.0–7.7		6.9	5.6–7.3	
Multiparous	21.1	6.6	6.3–6.9		6.4	6.0–6.9	
**Child characteristics**							
Sex				0.481			
Male	17.3	7.0	6.7–7.4				
Female	19.4	6.9	6.5–7.2				
Primary milk feeding method at 12 months				<0.001			<0.001
Breast milk	41.3	4.8	4.5–5.2		4.4	4.0–5.0	
Mixed-breast milk and formula	11.8	8.0	7.3–8.7		7.5	6.7–8.2	
Formula	2.3	9.2	8.9–9.5		9.2	8.8–9.6	
Neither breast milk nor formula	20.1	5.6	5.3–6.0		5.6	5.1–6.1	

^(a)^ EAR Estimated average requirement. ^(b)^ IRSAD, Index of Relative Socio-Economic Advantage and Disadvantage, where decile 1 = most disadvantaged and decile 10 = most advantaged. ^(c)^ BMI Body Mass Index.

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
