# Peer review of "Determinants and Sources of Iron Intakes of Australian Toddlers: Findings from the SMILE Cohort Study"

_ijerph, 2019, doi:10.3390/ijerph16020181_

Round 1

Reviewer 1 Report

A well written paper which fills a significant knowledge gap in literature. As there seems to be a dearth of research on iron intake of Australian toddlers. This is of significant importance as iron is needed by babies and children for normal brain development.

It was especially pleasant to see how the lower iron intake of breastfeeding toddlers in comparison to toddlers on formula was discussed without compromising the integrity of the benefits of continued breastfeeding. This was succinctly addressed in the second to the last paragraph of page 13 where the authors wrote the following sentences:

“Although children who received breast milk as their primary milk feed were more likely to have usual intakes below the EAR, breastfeeding to 12 months and beyond has been proven to provide numerous health benefits to the child and mother [42,43]. As such, it should continue to be championed and strongly promoted. However, it appears that other important messages about infant and toddler feeding are being missed by parents, particularly those relating to iron. Dwyer [44] on reviewing the findings of the 2016 wave of the US FITS study suggested that stronger recommendations are needed so that parents understand “the specific foods children should be eating and the developmentally appropriate times to introduce complementary foods and beverages” (p1578S). This appears also to be the case for Australian parents, particularly with regard to the introduction of iron-rich foods. While the Australian IFG make specific and strong recommendations with regards the introduction of iron-rich foods as first foods [37], the message does not appear to be getting through to parents.”

There was an emphasis on introduction of iron rich foods and the need to get the important message on toddler feeding across to parents. This point was also reinforced in the conclusion on page 14.

Author Response

Thank you for your positive response to our paper.

No changes were requested by this reviewer.

Reviewer 2 Report

This is an interesting and well written manuscript describing the iron intake of Australian toddlers.  Not-withstanding the problems of assessing dietary intake via self-report, the results indicate that a significant number of toddles are at risk of inadequate intake.

As not every reader will be familiar with the Multiple Source Method it would be helpful to add in the abstract that the MSM was used to combine data from recalls and food records to estimate usual iron intake.

Please clarify whether the 24 hr recalls were performed over the phone or face-to-face.

I was not aware of the MSM method and checked the website.  Seems it is mainly designed for combining FFQ and 24 hr recall data where the FFQ gives information on the chance that something will be consumed and the recall give more detail.  There are also choices to be made in the program and the methods could give more detail about these and how well the method works when used with recalls and records.

Liver is a rarely consumed and high iron food.  If it was not consumed on recording days a significant contribution to iron intake could be missed.  Is there any information on likely consumption?

Formatting of references is not consistent.  Journal titles should all be abbreviated or not depending on instructions to authors.

Line 13 in abstract, change ‘will’ to ‘may’ or ‘can’.

Remove brackets around ‘n’s in table 1 and put a footnote marker in table re BMI.

Author Response

This is an interesting and well written manuscript describing the iron intake of Australian toddlers.  Not-withstanding the problems of assessing dietary intake via self-report, the results indicate that a significant number of toddles are at risk of inadequate intake.

Response: Thank you for your overall support of the paper and your constructive feedback.

1. As not every reader will be familiar with the Multiple Source Method it would be helpful to add in the abstract that the MSM was used to combine data from recalls and food records to estimate usual iron intake.

Response: The abstract and the methods section have been revised to make it explicit that the data being used in the MSM came from the 24-h recall and the food records (See lines 19 and 99).

2. Please clarify whether the 24 hr recalls were performed over the phone or face-to-face.

Response: The 24-hr recalls were performed by telephone interview and this has been clarified in the methods section (line 79).

3. I was not aware of the MSM method and checked the website.  Seems it is mainly designed for combining FFQ and 24 hr recall data where the FFQ gives information on the chance that something will be consumed and the recall give more detail. There are also choices to be made in the program and the methods could give more detail about these and how well the method works when used with recalls and records.

Response: The MSM can be used with or without data from a Food Frequency questionnaire. When data from a FFQ is not available and therefore data on habitual consumption of a food is lacking then the default setting within program assumption option three, specifying 50% of those not having consumed the food at the 24-hour recalls as habitual consumers can be used  .  The Multiple Source Method has been used previously by researchers (Spence AC, et al.  J Acad Nutr Diet. 2018;118(9):1634-43 e1) to determine the usual intake of vegetables and fruits from data collected from three 24 hour recalls with each recall period being a ‘source’ of data. Similarly, in the current study each day was considered to be a different ‘source’ of data.

In the case of nutrients that are consumed daily then only the part of the model for consumption day amount is run. (Souverein OW, Dekkers AL, Geelen A, Haubrock J, de Vries JH, Ocke MC, et al. Comparing four methods to estimate usual intake distributions. Eur J Clin Nutr. 2011;65 Suppl 1:S92-101.) As iron was consumed on all days by all participants then consumption day amount part of the model was run. This detail  and further detail on the MSM has been added to the methods section (Lines 101-106).

4. Liver is a rarely consumed and high iron food. If it was not consumed on recording days a significant contribution to iron intake could be missed.  Is there any information on likely consumption?

Response: Indeed, liver is a significant source of iron but these days in Australia is rarely eaten.  Of the 828 children, only 2 children consumed liver in the form of pate/ liverwurst on any day.  In each case, the amounts consumed were very small (e.g. 1-2 teaspoons - probably due to the strong taste) and did not make a major contribution to their overall iron intake. While this is interesting, due to the extremely small number of children (0.2%) consuming liver we do not think that this warrants mention in the paper.

5. Formatting of references is not consistent.  Journal titles should all be abbreviated or not depending on instructions to authors.

Response: All journal titles have been abbreviated.

6. Line 13 in abstract, change ‘will’ to ‘may’ or ‘can’.

Response: The word ‘will’ has been changed to ‘can’ in the abstract and introduction.

7. Remove brackets around ‘n’s in table 1 and put a footnote marker in table re BMI.

Response: Brackets have been removed and footnote marker has been added